# Intramolecular and Metal-to-Molecule Charge Transfer Electronic Resonances in the Surface-Enhanced Raman Scattering of 1,4-Bis((*E*)-2-(pyridin-4-yl)vinyl)naphthalene

**DOI:** 10.3390/molecules24244622

**Published:** 2019-12-17

**Authors:** Isabel López-Tocón, Elizabeth Imbarack, Juan Soto, Santiago Sanchez-Cortes, Patricio Leyton, Juan Carlos Otero

**Affiliations:** 1Andalucía Tech, Unidad Asociada IEM-CSIC, Departamento de Química Física, Facultad de Ciencias, Universidad de Málaga, E-29071 Málaga, Spain; soto@uma.es; 2Instituto de Química, Pontificia Universidad Católica de Valparaiso, 2373223 Valparaiso, Chile; eimbarack@yahoo.es (E.I.); patricio.leyton@pucv.cl (P.L.); 3Instituto de Estructura de la Materia, Consejo Superior de Investigaciones Científicas, E-28006 Madrid, Spain; s.sanchez.cortes@csic.es

**Keywords:** SERS, resonance Raman, computational spectra, charge transfer, DFT calculations

## Abstract

Electrochemical surface-enhanced Raman scattering (SERS) of the cruciform system 1,4-bis((*E*)-2-(pyridin-4-yl)vinyl)naphthalene (bpyvn) was recorded on nanostructured silver surfaces at different electrode potentials by using excitation laser lines of 785 and 514.5 nm. SERS relative intensities were analyzed on the basis of the resonance Raman vibronic theory with the help of DFT calculations. The comparison between the experimental and the computed resonance Raman spectra calculated for the first five electronic states of the Ag_2_-bpyvn surface complex model points out that the selective enhancement of the SERS band recorded at about 1600 cm^−1^, under 785 nm excitation, is due to a resonant Raman process involving a photoexcited metal-to-molecule charge transfer state of the complex, while the enhancement of the 1570 cm^−1^ band using 514.5 nm excitation is due to an intramolecular π→π* electronic transition localized in the naphthalenyl framework, resulting in a case of surface-enhanced resonance Raman spectrum (SERRS). Thus, the enhancement of the SERS bands of bpyvn is controlled by a general chemical enhancement mechanism in which different resonance processes of the overall electronic structure of the metal-molecule system are involved.

## 1. Introduction

The analysis of the selective enhancement of particular bands recorded in surface-enhanced Raman scattering (SERS) of most systems like aromatic molecules is usually a very difficult task. These changes between the relative intensities of the Raman and the SERS spectra are due to the subtle effect of the nanostructured metal surface on the electronic structure of the molecule and imply a modification of the Raman selection rules. Nowadays, it is widely recognized that they are two main contributions responsible for the SERS enhancement based on a physical and a chemical mechanism, respectively [1,2], both of them being related to different types of resonant processes. The electromagnetic mechanism is the main responsible for the enormous enhancement of the electromagnetic field on the surface of the metal nanoparticle and dominates the SERS intensity in any intense experiment. This physical plasmonic mechanism allows for using the SERS as an ultrasensitive sensor for trace detection [3,4,5]. The chemical enhancement mechanism, in turn, can contain different contributions: (1) the effect of the adsorption of the molecule on the metal [6], which modifies the molecular properties in its electronic ground state, and/or (2) the presence of resonant Raman processes up to (2a) excited electronic states of the own adsorbate (surface-enhanced resonance Raman scattering (SERRS) [7] similar to resonance Raman of the molecule) or (2b) to new metal-to-molecule charge transfer states of the surface complex (SERS-CT) [8,9,10,11], where an electron is transferred in the transient excited state. While the metal plays the main role in the electromagnetic mechanism, the chemical contributions are very dependent on the particular molecular system and the experimental conditions. This is the reason why it is very difficult to know which processes are acting in each SERS and, therefore, to quantify the relative participation of each of them. This work continues a series of studies where we demonstrated the close relationship between the electronic structure of the metal-molecule complex and the SERS spectra, particularly concerning the selective enhancements of the bands. These studies show how more or less simple theoretical calculations are able to account for the complex behavior of this kind of spectra. We developed a systematic methodology to predict the effect of resonant Raman processes in the SERS relative intensities that allows to detect the presence of resonances in particular SERS experiments.

To elucidate which mechanisms contribute to the selective enhancement of the bands in a SERS record, it is necessary to resort to quantum chemical calculations of a model of the metal-molecule surface complex. Although different strategies can be proposed depending on the level of the calculations or the selected model of the metal-adsorbate supermolecule [12,13,14], it is very complicated to take into account the effect of the electrode potential in the calculations. The electrode potential tunes the overall electronic structure of the surface complex, which determines the strength of the molecular adsorption as well as the energies of the perturbed intramolecular excited states and in particular, the metal-molecule charge transfer states. Our proposal of analysis is based on DFT electronic structure calculations for selected molecule-metal clusters [15,16,17]. Thereafter, the theoretical electronic spectra [18,19,20] computed on the basis of the resonance Raman vibronic theory are compared with the experimental SERS in order to detect the presence of resonant Raman processes of different types. This methodology has proven its powerfulness to explain the complex dependence of the spectra on the applied potential and the differentiated behaviours shown by the SERS of different molecular systems.

In this work, SERS spectra of 1,4-bis((*E*)-2-(pyridin-4-yl)vinyl)naphthalene (bpyvn) recorded on nanostructured silver surface at different exciting laser lines and electrode potentials were analyzed with the mentioned strategy. CAM-B3LYP/def2-TZVPP DFT calculations of a simple molecular model of the metal–bpyvn surface complex were carried out, where the metal was simulated by mean of two silver neutral atoms (Ag_2_–bpyvn) bonded to the molecule through the pyridyl nitrogen atom. The Bpyvn molecule belongs to a pyridine-appended π-conjugated family used as a building block in luminescent coordination polymers [21] whose chemical and physical properties are tuned by modifying different parameters as, for instance, the π-conjugated groups of the organic ligands. Moreover, bpyvn is a complex system which can be used as pyridine-based linkers in DNA bis-intercalators to probe the spatial organization of DNA [22] and it is a potential candidate to be used as a molecular junction in electronic devices as happens is other related molecules [23,24]. This cruciform adsorbate is a bifunctional system able to bond to two metallic electrodes of a molecular junction through the nitrogen atoms of the two opposite pyridyl rings. The electrical conductance of the junctions will be controlled by the electronic structure of the metal-molecule complex and, in particular, by the metal-molecule charge transfer states whose existence and properties will be studied in this work by means of surface-enhanced Raman scattering.

## 2. Results and Discussion

### 2.1. Electrochemical SERS Spectra

Raman spectrum of purified solid bpyvn was recorded at 1064 nm excitation to avoid fluorescence emission (Figure 1). The spectrum is characterized by three bands in the 1500–1700 cm^−1^ region and the vibrational assignment was carried out on the basis of the force field results. Two very strong lines were recorded at 1625 and 1575 cm^−1^ and are assigned to the stretching of C=C vinylene bonds and to the 8a mode of naphthalene (naph), respectively, while the medium intensity band recorded at 1592 cm^−1^ correspond to the 8a mode of pyridines (py). Other three medium/strong bands were recorded at 1368, 1337 and 1202 cm^−1^, respectively. The first of them is an in-plane deformation of naphthalene, (δ(CH)naph) and the two last ones are located in the pyridine rings being assigned to the normal modes 14; ν (CC)py and 9a; δ(CH)py. Two medium bands are registered near 1000 cm^−1^ which are assigned to the ring-deformation mode 12; δring of pyridine (988 cm^−1^) and to a complex vibration involving out-of-plane CH deformations of pyridine and naphthalene, γ(CH)py,naph, respectively (969 cm^−1^). The experimental and calculated vibrational wavenumbers of these main Raman bands are summarized in Appendix A as well as the assignment made from the calculated DFT modes visualized with the MOLDEN [25] program. No Raman spectrum of the aqueous solution could be recorded due to the very poor solubility of the sample.

Electrochemical SERS of bpyvn recorded at different electrode potentials by using 785 nm and 514.5 nm can be compared in Figure 2. It can be seen that both series of spectra show the same main bands as those recorded in the Raman spectrum. No bands due to original reagents used in the synthesis or subproducts of other chemical reactions were detected, even at the low concentration of sample in the spectroscopic experiments (10^−4^ M).

Appendix A contain the dependence of the experimental SERS wavenumbers on the electrode potential using 785 nm and 514.5 nm excitation lines, respectively. In both cases, there is no significant wavenumber shifts from 0 up to −1.00 V indicating that no molecular reorientation occurred. Concerning the SERS intensities and on the basis of the propensity rules of electromagnetic mechanism [26,27], a perpendicular molecular orientation of the adsorbate was concluded with respect to the metallic surface given that the strongest SERS bands correspond to in-plane normal modes. The interaction between the metal and the molecule should be through the nitrogen atom of one of the two pyridyl moieties as occurs in pyridine [28] and similar derivatives [8,9,10,11]. This is the reason why the Raman bands of modes 8a and 12 of pyridine recorded at 1592 and 988 cm^−1^ show the largest blue shifts in SERS with respect to the normal Raman spectrum.

The relative intensities of the SERS bands are not very sensitive to the electrode potential. However, significant changes can be found in the 1500–1600 cm^−1^ region when the two series of SERS and the normal Raman spectrum are compared. In the case of 785 nm excitation, the SERS bands recorded at about 1625 and 1574 cm^−1^ and assigned to ν(C-C) and 8a; ν (CC)naph, respectively, show a very similar relative intensity as occurs in the normal Raman, but the band at about 1607 cm^−1^ shows a relative enhancement, which is potentially dependent, becoming as strong as the two previous ones in the spectrum recorded at −0.6 V, for instance. Therefore, the most striking feature of this series of SERS registered under 785 nm excitation is the selective enhancement of this last vibration 8a; ν (CC)py which is located in the pyridine moieties. Other bands assigned to pyridyl groups also show relative enhancement but of a lower intensity, such as those recorded at 1337 and 1203 cm^−1^. We have shown in previous works that this behavior is related to the presence of a metal-to-molecule resonant charge transfer mechanism in the SERS of aromatic molecules [8,9,10,11,29,30,31].

The SERS recorded using the more energetic 514.5 nm excitation line shows a different behavior. These spectra are characterized by the strong and selective enhancement of the band recorded at 1575 cm^−1^ corresponding to the 8a; ν (CC)naph fundamental, which is the strongest one at any electrode potential.

In summary, different behaviours were detected in the SERS of bpyvn adsorbed on a nanostructured silver surface depending on the excitation lines. At lower excitation energy (785 nm, 1.6 eV), the SERS band assigned to the very characteristic 8a normal mode located in the pyridyl framework was enhanced, while a normal mode assigned to the naphthalenyl group was selectively enhanced when the more energetic green line was used (514.5 nm, 2.4 eV). The next step will be devoted to relating the effect of the electrode potential and the energy of the laser line to the electronic structure of the metal–molecule system by using DFT calculations [20].

### 2.2. Electronic Structure of bpyvn and Ag_2_-bpyvn Complex

The optimized structures (Appendix A) and the respective force fields of isolated bpyvn and the Ag_2_-bpyvn complex in their respective ground electronic states (S_0_) were carried out at the same CAM-B3LYP/def2-TZVPP level of theory previously used in a related system [7]. A neutral silver dimer attached to bpyvn through one of its nitrogen atoms was selected as a model of the surface complex near the potential of zero charge of a rough silver electrode which would lie in the range −0.5 V to −0.8 V [32]. The Gibbs free energy for the formation of the complex was calculated according to the Statistical Thermodynamics [33,34] and amounts to −15 kcal/mol and the energy profile of the formation of the complex was computed with the linear interpolation method [35,36,37,38,39] (Appendix A).

Table 1 shows the time-dependent vertical energies and the corresponding oscillator strengths of the first ten singlets of bpyvn and Ag_2_-bpyvn. The first electronic transition up to the S_1_ state of isolated bpyvn was calculated at 3.6 eV and shows the largest oscillator strength of 1.11. Therefore, this state could be in resonance or preresonance conditions when it is excited using the more energetic 514.5 nm (2.4 eV) laser line. This could be the cause of the intense fluorescence emitted for this molecule which prevents recording the Raman spectra, making it necessary to resort to the less energetic line of 1064 nm (Figure 1).

Regarding the Ag_2_-bpyvn complex, the Mulliken charge corresponding to the Ag_2_ cluster and the transferred charge (∆q) from Ag_2_ to bpyvn in the corresponding S_i_ states are also summarized in Table 1. The first excited electronic state of the complex corresponds to a CT state (S_1(CT0)_) given that a transferred charge of −0.80 a.u. was calculated. This CT_0_ state has a lower vertical energy (2.6 eV) than that S_1_ state of the isolated molecule (3.6 eV), which is calculated at 3.2 eV in this case (S_2_). The difference between the energies of S_1(CT0)_ and S_2_ or S_3_ amounts ca. 1 eV and, therefore, is very similar to that of the two employed laser lines (0.8 eV). Other more energetic CT states of the complex (S_4(CT1)**,**_ S_5(CT2)_ and S_10(CT3)_) are calculated at energies of 3.8, 3.8 and 4.6 eV. If the S_1_ state of bpyvn had been reached with the 514.5 nm laser line, the SERS recorded under 785 nm excitation could have been in preresonance or resonance with the first CT state. In the next section, the theoretical Raman spectra of the Ag_2_-bpyvn complex in preresonance with the first five electronic transitions will be computed according to the Albretch theory [40,41].

### 2.3. Theoretical Resonance Raman Spectra

Figure 3a shows the theoretical Raman spectra in preresonance for the main electronic transitions of bpyvn and Ag_2_-bpyvn calculated within the vibronic approximation of the Albretch theory (see Section 4.2.2). All the bands are located at their computed wavenumbers without numerical scaling. At the bottom of Figure 3a, one can see the resonance Raman spectrum for the first S_1_–S_0_ electronic transition of bpyvn which can be compared to those of the five electronic transitions of the Ag_2_-bpyvn complex. All calculated spectra were convoluted with a Lorentzian function and half width at half maximum of 5 cm^−1^. The main molecular orbitals involved in the corresponding electronic transitions are also shown on the right in Figure 3b.

In addition to the two medium bands calculated at 1733 and 1405 cm^−1^ and assigned as ν(C=C) and δ(CH)naph, respectively, the S_1_–S_0_ theoretical spectrum of bpyvn is characterized by the very strong band calculated at 1659 cm^−1^ which correspond to that observed at 1575 cm^−1^ and assigned to the 8a; ν (CC)naph fundamental. The molecular orbitals involved in this electronic transition indicate a HOMO-LUMO, π–π* transition localized in the naphthalenyl framework. This pure intramolecular electronic transition is also found in the Ag_2_-bpyvn complex where it appears mixed, contributing in the two S_2_–S_0_ and S_3_–S_0_ transitions with similar energies and oscillator strengths. The spectra calculated in resonance with these two states of the hybrid system are very similar to that of the isolated molecule and are also dominated by the strong intensity of the same band. Therefore, the selective SERS enhancement of the 8a; ν (CC)naph vibration recorded at 1575 cm^−1^ using the 514.5 nm laser line is due to an intramolecular π–π* transition located in the naphthalenyl framework and these spectra can be called SERRS.

However, the calculated spectrum for first metal-to-molecule CT electronic transition of the complex (S_1(CT0)_–S_0_) predicts the selective enhancement of the 1675 cm^−1^ band assigned to the 8a; ν (CC)py fundamental located in the pyridyl framework, just as it occurs in the SERS spectra recorded by exciting with the 785 nm laser line. This means that the transferred electron is temporarily accommodated in the pyridyl ring closer to the metal surface, as can be seen in the corresponding Figure 3b. The transitory species deforms the geometry of the adsorbate along the 8a mode of pyridine, explaining the enhancement of this band that is in resonance with the first CT state, and allowing these spectra to be called SERS-CT. The energy of this CT transition could be tuned by the electrode potential and this is the reason why the intensity of this band is much more dependent on the applied potential than the remaining ones.

The other two S_4(CT1)_-S_0_ and S_5(CT2)_-S_0_ CT transitions of the complex have similar energies to the strong intramolecular S_2_-S_0_ and S_3_-S_0_ excitations, but their role in the SERS spectra recorded with the 514.5 nm line should be minor. Both CT states are close in energy (3.8 eV) to that of pure intramolecular S_2_/S_3_ excited states (3.2/3.5 eV) and could be reached under the same experimental conditions but the calculated oscillator strengths are not significant (0.002 and 0.007, respectively) when compared to the values of 1.2 and 0.7 calculated in the case of the S_2_ or S_3_ states.

## 3. Conclusions

The different Raman responses of bpyvn adsorbed on rough silver electrode when irradiated by the two different excitation laser lines, 785 nm and 514.5 nm, can be explained on the basis of the electronic structure of the hybrid metal-molecule system. The DFT computational spectra calculated according to a modified vibronic theory of resonance Raman for the simple model of surface complex, Ag_2_-bpyvn, points out that the enhancement of the bands recorded at about 1605 and 1574 cm^−1^, and assigned to 8a; ν (CC)py and 8a; ν (CC)naph modes, respectively, is controlled by a general chemical enhancement mechanism acting in the SERS experiments through different resonant Raman processes involving electronic states of different nature. A photoexcited metal-to-molecule charge transfer state is responsible for the selective enhancement of the band at 1605 cm^−1^ under 785 nm excitation (SERS-CT), while a pure intramolecular excited state of the adsorbate is involved in the case of the enhancement of the band at 1574 cm^−1^ using the 514.5 nm laser line. Therefore, in this last case, the spectra correspond to a surface-enhanced resonance Raman (SERRS) process.

The reported results support the usefulness of SERS in studying the subtle electronic structure of charged interfaces and the close relationship between the electronic structure of the metal–molecule surface complex and the SERS. Once again, our methodology demonstrated to be a very useful tool to predict the effect of resonant processes in the SERS of aromatic molecules, being also able to explain the differences found between the differentiated behaviors observed in the spectra of related molecular systems.

## 4. Materials and Methods

### 4.1. Experimental Section

#### 4.1.1. Synthesis of 1,4-bis((*E*)-2-(pyridin-4-yl)vinyl)naphthalene, (bpyvn)

The synthesis of bpyvn was made by means of a Heck C–C coupling [42,43]. The procedure is quite similar to that previously published [21] but with some differences. In a two-neck balloon, 200 mg (0.69 mmol) of the reagent 1,4-dibromonaphthalene, 15 mg of palladium acetate and 130 mg of the tri-o-tolylphosphine reagent was added. The balloon was connected to a reflux system and sealed in an inert atmosphere. Subsequently, 6 mL of dry tetrahydrofuran and 3 mL of triethylamine were added and the mixture was heated to 67 °C. Finally, 0.4 mL (3.71 mmol) of 4-vinylpyridine was added in 4 portions of 100 μL every 20 min.

The reaction was maintained with constant stirring at reflux temperature for 36 h, then it was poured into water and extracted with dichloromethane (3 × 10 mL). The organic phase was dried with calcium chloride and then filtered and concentrated in rotavapor. The crude of this reaction was purified by column chromatography using silica gel 60 (70–230 mesh) as the stationary phase and a 3:1 ethyl acetate/dichloromethane solvent mixture was used as the mobile phase. The reaction yield obtained after all purification processes was 41% yield. The color of purified bpyvn was a little bit yellowish. Bpyvn melting point = 234 ˚C. RMN^1^H (CDCl_3_, 300MHz, δ (ppm)): 8.66 (d, 4H, *J* = 5.97 Hz); 8.27 (m, 2H); 8.13 (d, 2H, *J* = 16.02 Hz); 7.83 (s, 2H); 7.65 (m, 2H); 7.49(d, 4H, *J* = 6.05 Hz); 7.13 (d, 2H, *J* = 16.02 Hz). RMN^13^C (CDCl_3_, 125 MHz, δ (ppm)): 150.37; 144.61; 134.65; 131.53; 130.00; 129.93; 129.41; 126.64; 124.20; 121.04.

#### 4.1.2. Electrochemical SERS Records

SERS spectra were recorded by using an InVia-Reflex micro-Raman spectrometer (Renishaw, Wotton-under-Edge, Gloucestershire, UK). The microscope was equipped with a macro objective (f:30 mm) and the spectral resolution was set to 2 cm^−1^. Two excitation laser lines were used at 785 nm and 514.5 nm wavelengths, with a laser power of 4.0 mW and 6.0 mW in the sample, respectively. Wire 2.0 software (Renishaw) implemented in the equipment was used for spectral acquisition and manipulation.

Electrochemical equipment composed of a potentiostat model 600E (CH Instruments Inc., Austin, TX, USA) and a non-commercial three electrodes cell was used to control the electrode potentials and to record SERS spectra. The electrodes are a platinum counter electrode, an Ag/AgCl/KCl(sat.) reference electrode and a pure silver working electrode which was polished with 1.00, 0.30 and 0.05 µm alumina (Buehler, Lake Bluff, IL, USA). The working electrode was electrochemically activated in order to produce the required SERS active nanostructures by maintaining the electrode potential at −0.5 V and then subjecting it to seven pulses at +0.6 V for 2 s. A 0.1 M aqueous solution of Na_2_SO_4_ was employed as electrolyte in the activation procedure. SERS were recorded from an aqueous solution of 0.1 M Na_2_SO_4_ and 10^−4^ M bpyvn. The water employed in all solutions was obtained in a Milli-Q system, 18.2 MΩ cm resistivity.

SERS spectra were sequentially recorded from 0.0 V up to −1.0 V by step of −0.1 V, with the accumulation of 1 scan and 10 s exposure time to avoid sample damage. The sequence of spectra was recorded three times and there were no significant fluctuations for a particular spectrum, indicating a good reproducibility of the results.

Raman spectra of the solid obtained at 1064 nm excitation laser line were recorded on an FT-Raman RFS 100/S spectrometer (Bruker, Billerica, MA, USA) provided with an Nd:YAG laser and a Ge detector cooled with liquid nitrogen. The laser power reaching the sample was about 200 mW, and the spectral resolution was 4 cm^−1^.

### 4.2. Computational Details

#### 4.2.1. DFT Calculations

The hybrid exchange-correlation functional, CAM-B3LYP [44], with the def2-TZVPP [45,46] basis sets was employed to calculate the optimized structures and the vibrational wavenumbers of bpyvn and the model of the surface complex, Ag_2_-bpyvn, in the ground electronic state, S_0_. Then, their respective electronic structures, that is, the vertical energy of the first ten excited electronic states at the Franck- Condon point (S_i = 1-10_) were calculated by using the time-dependent DFT method at the same level of computation. This type of calculation has already been successfully employed in a previous work to predict the electronic structure of a related molecule [7].

Different conformers of bpyvn can be optimized, depending on the relative position of the pyridine planes with respect to the naphthalene one. We chose the most stable structure by rotating both aromatic rings with respect to the naphthalene [7]. Two silver atoms linked to the molecule through one of the nitrogen atom were added to optimize the Ag_2_-bpyvn complex. All calculated vibrational wavenumbers of bpyvn and Ag_2_-bpyvn are real, indicating that they correspond to minima.

All DFT calculations were carried out with the GAUSSIAN16 program [47]. The molecular orbitals and the vibrational wavenumbers were analyzed with the help of the visualization MOLDEN program [25].

#### 4.2.2. Computational Resonance Raman Spectra

Resonance Raman spectra of bpyvn and Ag_2_-bpyvn complex were calculated according to a modified vibronic theory of Albrecht [40,41], in which only Franck-Condon factors are taken into account. Details of the calculations of the resonance Raman intensities have been given previously [7,48,49,50,51,52]. Briefly, the method follows the independent mode displaced harmonic oscillator (IMDHO) approach where small displacements of the excited state with respect to the ground state geometry are considered to be proportional to the gradient (forces) on the excited state potential energy surface. No rotation in the electronic states involved in the resonance process was assumed. For detailed information, see Appendix A.

## Figures and Tables

**Figure 1 molecules-24-04622-f001:**
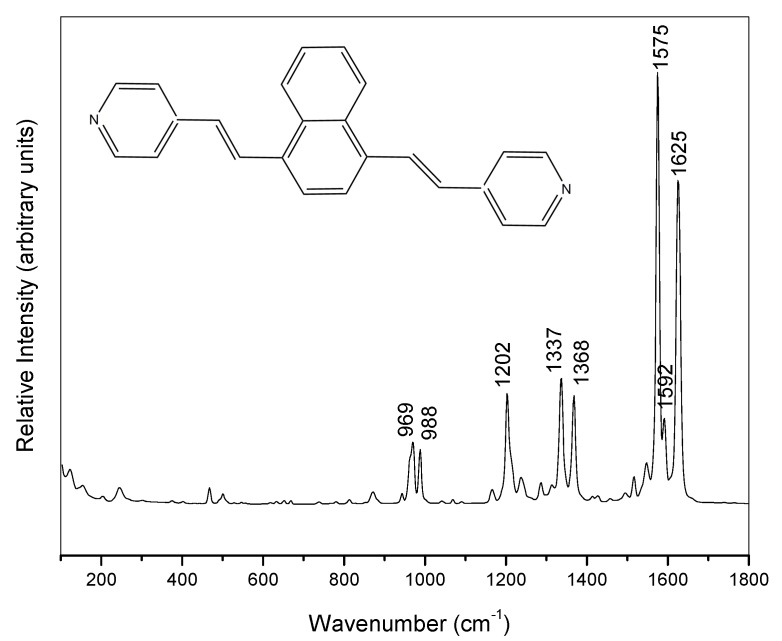
Raman spectrum of solid 1,4-Bis((*E*)-2-(pyridin-4-yl)vinyl)naphthalene (bpyvn, inset) (1064 nm excitation).

**Figure 2 molecules-24-04622-f002:**
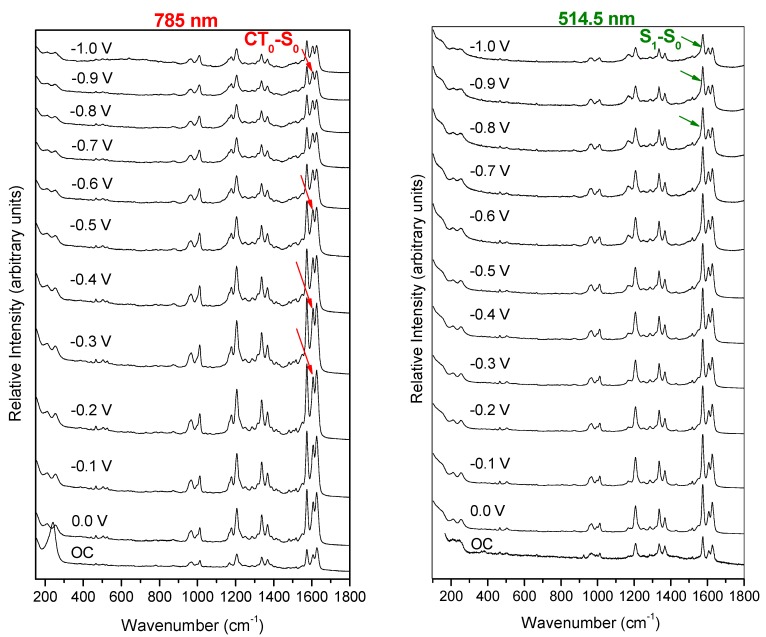
SERS spectra of bpyvn/Na_2_SO_4_ (10^−4^ M/0.1 M) recorded on silver at different electrode potentials ranging from 0.0 V up to −1.0 V using the excitation lines of 785 nm (left) and 514.5 nm (right), respectively. Electrode potentials are measured vs. the Ag/AgCl/KCl(sat.) reference electrode. The bottom spectra correspond to open-circuit voltage (OC).

**Figure 3 molecules-24-04622-f003:**
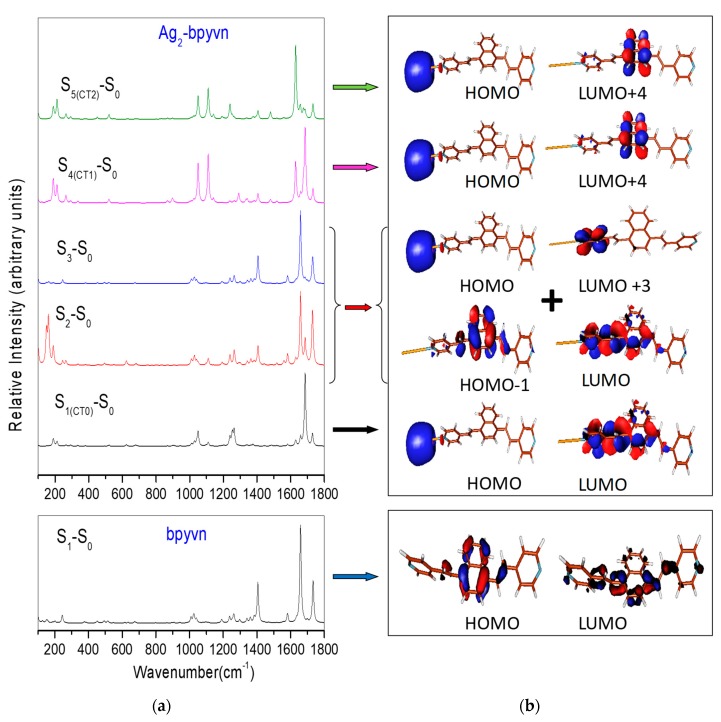
(**a**) Calculated resonance Raman spectra for the first five S_i_-S_0_ electronic excitations of the Ag_2_-bpyvn complex and theoretical spectrum of the first electronic transition S_1_-S_0_ of bpyvn. (**b**) Relevant molecular orbitals involved in the corresponding electronic transitions of Ag_2_-bpyvn and bpyvn depicted in Figure 2a.

**Table 1 molecules-24-04622-t001:** CAM-B3LYP/def2-TZVPP excited singlet states of bpyvn and Ag_2_-bpyvn calculated at the Franck-Condon point.

		S_1_	S_2_	S_3_	S_4_	S_5_	S_6_	S_7_	S_8_	S_9_	S_10_
**bpyvn**	E (eV)	3.590	4.369	4.628	4.747	4.765	4.789	5.047	5.087	5.094	5.113
	f ^1^	1.113	0.000	0.102	0.017	0.017	0.160	0.194	0.015	0.023	0.031
		**S_1(CT0)_**	**S_2_**	**S_3_**	**S_4(CT1)_**	**S_5(CT2)_**	**S_6_**	**S_7_**	**S_8_**	**S_9_**	**S_10(CT3)_**
**Ag_2_-bpyvn**	E (eV)	2.611	3.237	3.535	3.808	3.824	4.049	4.160	4.348	4.505	4.604
	f ^1^	0.001	1.150	0.680	0.002	0.007	0.274	0.289	0.001	0.090	0.010
	Ag_2_ charge ^2^	0.59	−0.23	−0.24	0.60	0.59	−0.23	−0.21	−0.21	−0.22	0.27
	∆q ^3^	−0.80	0.02	0.03	−0.81	−0.80	0.02	0.00	0.00	0.01	−0.48

^1^ Oscillator strength for the S_i_-S_0_ transitions. ^2^ Mulliken charge corresponding to the Ag_2_ cluster. ^3^Transferred charge from Ag_2_ to bpyvn in the S_i_ states of the complex. (∆q = qAg_2_(S_0_) − qAg_2_(S_i_)).

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
