# Peer review of "Intramolecular and Metal-to-Molecule Charge Transfer Electronic Resonances in the Surface-Enhanced Raman Scattering of 1,4-Bis((E)-2-(pyridin-4-yl)vinyl)naphthalene"

_molecules, 2019, doi:10.3390/molecules24244622_

Round 1

Reviewer 1 Report

Comments on manuscript 664655 submitted to the Molecules

Title: “Intramolecular and metal-to-molecule charge transfer electronic resonances in the surface-enhanced Raman scattering of 1,4-Bis((E)-2-(pyridin-4-yl)vinyl)naphthalene”

Authors: I. López-Tocón, E. Imbarack, J. Soto, S. Sanchez-Cortes, P. Leyton, J.C. Otero

The paper deals with an interesting surface-enhanced Raman spectroscopy (SERS) response of the title molecule, modulated by different electrode potentials applied to an activated silver surface and different laser lines used to record the spectra. The quality of the data presented is very good and the discussion performed by the authors is clear and deeply explores the experimental results. Particular remarkable is the ability of the DFT electronic structure calculations followed by the computation of electronic spectra, using the resonance Raman vibronic theory, to reproduce the observed selective enhancements of the SERS signals under both different applied potential and excitation lasers allowing to unveil the subjacent mechanisms.

In my opinion, this is a thorough and accurate study that should be published in Molecules, after addressing the following issues:

The authors indiscriminately use “respectively”, too often along the text and in some cases (g. lines 19, 96, 101, 161, 174, 273) unnecessarily. Lines 42 to 48: confusing, please clarify. Lines 131 to 133: confusing, please clarify. Self-citations must be limited (32 out of 53 is too much). As a new synthetic route was used and the compound melting point differs 5 ºC from the reported one [21] (234 vs. 239 ºC); hence, the elemental analysis should be included. The mathematical expression for calculating the intensity of the Raman bands (and related equations, Section 4.2.2) should be presented as Supplementary Material.

Editing typos:

Line 45: “thee” should be “the” Line 46: “adsorbte” should be “adsorbate” Line 94: “bond” should be “bonds” Line 95: “pyridine” should be “pyridines” Line 97: “ring” should be “rings” Line 135: “moiety” should be “moieties” Line 135: “group” should be “groups” Line 145: “1.5 eV” should be “1.6 eV” Line 147: “2.3 eV” should be “2.4 eV” (as in line 164) Lines 162 & 171: “3.5 eV” should be “3.6 eV” Line 216: “… CT transition should be have to be tuned …”. Which of the two? “should” or “have to” Line 403: Format reference

Reviewer 2 Report

The manuscript written by López-Tocón et al., entitled “Intramolecular and metal-to-molecule charge transfer electronic resonances in the surface-enhanced Raman scattering of 1,4-Bis((E)-2-pyridin-4-yl)vinyl)naphthalene” (Manuscript ID: molecules-664655), experimentally analyzed the solid and surfaced-enhanced Raman spectra and electrode potentials of bpyvn and theoretically simulated the ones of a truncated model (Ag2-bpyvn) using density functional theory. The authors concluded that excitation with the longer wave length of 785 nm induces the enhancement through charge transfer from Ag to bpyvn, while that with lower wave length does through intramolecular π–π* excitation. From theoretician’s point of view (this reviewer is not familiar with experiment), the results are presented and discussed in a nice shape, and the manuscript should be acceptable for publication in the present form. The following minor points may be addressed optionally.

One thing which is not clear to me is the lines 151 to 153. Did the authors perform force field, i.e. molecular mechanics, calculations? The present sentence implies that the authors performed MM calculations as well as DFT ones.

Lines 216 to 218 are very strange.

In the caption of Figure S2, the distance should be d(Ag–N)

Reviewer 3 Report

This manuscript describes the value of SERS in studying the subtle electronic structure of charged interfaces and the close relationship between the electronic structure of the metal–molecule surface complex and the SERS. This is a valuable tool to predict the effect of resonant processes in the SERS of aromatic molecules and potential applications as probes.

According to this reviewer opinion, this is an excellent manuscript that represents a novel concept and I support the publication of this manuscript in Molecules as it is
